# Stretchable and Low-Haze Ag-Nanowire-Network 2-D Films Embedded into a Cross-linked Polydimethylsiloxane Elastomer

**DOI:** 10.3390/nano9040576

**Published:** 2019-04-09

**Authors:** Ki-Wook Lee, Yong-Hoe Kim, Wen Xuan Du, Jin-Yeol Kim

**Affiliations:** School of Advanced Materials Engineering, Kookmin University, Seoul 136-702, Korea; Kwlee7908@kookmin.ac.kr (K.-W.L.); 20091304@kookmin.ac.kr (Y.-H.K.); duwenxuan1314@naver.com (W.X.D.)

**Keywords:** 15-nm silver nanowire, Pressure-induced polyol method, stretchable transparent electrode 2-D films, Low-haze, Embedded electrode film

## Abstract

We report the fabrication of stretchable transparent electrode films (STEF) using 15-nm-diameter Ag nanowires networks embedded into a cross-linked polydimethylsiloxane elastomer. 15-nm-diameter Ag NWs with a high aspect ratio (˃1000) were synthesized through pressure-induced polyol synthesis in the presence of AgCl particles with KBr. These Ag NW network-based STEF exhibited considerably low haze values (<1.5%) with a transparency of 90% despite the low sheet resistance of 20 Ω/sq. The STEF exhibited an outstanding mechanical elasticity of up to 20% and no visible change occurred in the sheet resistance after 100 cycles at a stretching-release test of 20%.

## 1. Introduction

Transparent conductive electrode films have increasingly attracted attention owing to their potential applications in optoelectronic fields, including touch screens, organic light emitting diodes and organic solar cells [1,2,3,4]. In particular, functionalized stretchable films, which have recently begun to be widely used as stretchable transparent electrode films (STEF), respond to mechanical deformations by the changes in electrical characteristics, such as resistance, owing to their stretchability and reproducibility. In this regard, nanomaterials, such as silver nanowires (Ag NWs) [5], single-walled carbon nanotubes (SWCNTs) [6,7,8], and graphene sheets [9,10,11], and their hybrid structures have been reported for use as sensitive strain sensors, which make them ideal for use as a transparent conductor in flexible or stretchable devices [12,13,14,15].

Among them, silver nanowires (Ag NW) have been gaining interest as a promising transparent conductive electrode material because of its simple synthesis and the possibility of large-area coating film fabrication via solution processes [16,17,18,19,20,21]. Particular attention has been focused on random network films of Ag NWs because such films can be easily fabricated in solutions and exhibit enhanced optoelectronic properties. The intrinsic properties of NWs mainly depend on the diameter and length of NWs. Many recent studies have also focused on the synthesis of Ag NWs with small diameters and large aspect ratios, which possess a low haze value due to low light scattering and good plasmonic properties. Polyol synthesis is known to be the most widely used and versatile method for the preparation of Ag NWs. To thin down the diameters, various polyol processes are being developed. In this regard, Wiley and a co-worker [3] recently reported the synthesis of Ag NWs with diameters of ~20 nm by controlling the bromide ion concentration in the conventional polyol method. Our group also have recently reported the synthesis of 20-nm-diameter Ag NWs under a pressure-induced polyol method in the presence of NaCl–KBr co-salts [22], but their synthesis mechanisms did not fully account for the synthesis. However, despite the progress, the synthesis of thin Ag NWs less than 20 nm has had limited success, meaning more research is required to synthesize wires with diameters below that amount. 

Herein, we report a novel pressure-induced polyol method for synthesizing ultra-thin Ag NWs with a diameter of 15-nm-diameter or less and a high aspect ratio (˃1000), a relatively unreported area so far. In particular, we have investigated the growth of Ag NWs and seed crystals in the presence of the AgCl-KBr co-salts instead of the NaCl salt used in the previous work [22] under a pressure of 1000 psi, and found that the K^+^ ions cause a remarkable pressure effect. 

For a two-dimensional (2-D) film consisting of an Ag NW networks, in particular, it has excellent transmittance and sheet resistance, yet its optical haze still needs to be improved in order for it to be suitable for display applications. Therefore, ultra-thin Ag NWs can be a good candidate for low-haze transparent electrodes. In particular, in order to obtain low-haze Ag NW network conductive films superior to indium tin oxide (ITO, up to 90% transmittance and ~1% haze at the low sheet resistance of 60 ohm/sq) in terms of opto-electrical performance, a diameter of at least 20 nm Ag NWs is required. However, to achieve the required optical characteristics, more effective processes that can control the shapes and sizes of the synthesized Ag NWs are required. In this work, 2-D films based on 15-nm-diameter Ag NW networks embedded into cross-linked polydimethylsiloxane (PDMS) elastomer, STEF, were formed via a conventional wet-coating technique that adhered the NWs to a PDMS substrate film, for flexible display applications, as shown in Figure 1. In particular, the conductor comprising an Ag NW network embedded into PDMS exhibited high elasticity, cycling stability, transparency, and excellent electrical conductivity. In addition, these films were also confirmed to exhibit good responses to the stretch/release for ≥100 cycles, while hysteresis tests without the loss of conductivity under stretching conditions of 20% were also conducted.

## 2. Results and Discussion

Herein, we newly synthesized the ultra-thin Ag NWs with 15 nm or less in diameter and aspect ratio to as high as 1000 using a pressure-induced polyol method via the chemical reduction of AgNO_3_ in the presence of an AgCl crystal and KBr (molar ratio = 2:1), according to a previous report [22]. Figure 2 is a plot of the change in diameter of Ag NWs synthesized at various pressure conditions (the four pressure values; 0, 110, 250, and 1000 psi). As shown in Figure 2(I), the diameter of the Ag NWs decreased with increasing pressure in the presence of KBr supplemented with AgCl, including NaCl and FeCl_3_ salts. In particular, at the highest reaction pressure (1000 psi (69 bar)), the Ag NWs that formed in the presence of AgCl with KBr were ultrathin with a mean diameter of 15 nm and a narrow size distribution (within ±5 nm). In any case, Ag NWs synthesized under the pressure-induced conditions of the present experiment were noticeably smaller and more evenly dispersed than those produced at atmospheric pressure. In contrast, in NaCl, AgCl, and FeCl_3_ supplemented with NaBr, the NW diameter was independent of pressure, as shown in Figure 2(II). These results suggest that in the presence of KBr, particularly in the presence of K^+^ ions, the pressure controls the rate of the formation of Ag^+^ ions, thereby suppressing the growth in the thickness direction of the wire. As a result, in the pressure-induced polyol reaction, the reduction in diameter of Ag NW was observed to be affected by pressure only in the presence of K^+^ ions. This suggests that K^+^ ion acts as an effect of pressure on the growth of Ag NW, but it was difficult to describe the kinetics of K^+^ ions involved in the formation and growth of Ag NWs in this work. However, in the process of synthesizing the Ag NWs, K^+^ ions greatly acted on the pressure, and ultra-fine Ag NWs with a diameter of 15 nm could be successfully synthesized. Liao et al. [23] explained that increasing the reaction pressure lowers the energy barrier of nucleation and accelerates nucleation, resulting in a controlled rate of metal nanostructure formation when pressure is applied. Here, the nucleation rate of Ag ions is also closely related to the wire size. Figure 2(III) shows the SEM images of the produced Ag NWs synthesized in the presence of AgCl–KBr salts; (a) 19–25, (b) 17–18, and (c) 15–16 nm, respectively. These NWs correspond with 0, 250, and 1000 psi (69 bar), respectively.

Figure 3 displays SEM images at low magnification of the 15-nm-diameter Ag NWs synthesized at 1000 psi. Subsequently, the small-size Ag seed particles grew into Ag NWs with a mean diameter of 15 nm (range: 6–20 nm; aspect ratio: ~800). The diameter distribution of the synthesized wires is plotted in Figure 3(II). The mean diameter is at least 5 nm smaller than that of NWs formed at 0 psi (mean diameter = 22 nm; distribution = 14–28 nm). The surface plasmon resonance (SPR) signals have inherent characteristics depending on the size and structure of the nanomaterials [4,24,25]. Therefore, the size of Ag NWs can be predicted from the absorption bands appearing at different frequencies critically in the SPR data. In this regard, the SPR characteristic peak of 15-nm-diameter Ag NW synthesized at 1000 psi pressure with AgCl-KBr present shows at 354 and 362 nm, as shown in Figure 3(V). The SPR peak in Figure 3(V) appeared at 362 nm, which was significantly shorter than those in wires with diameters of 20–22 nm [366-nm peak; see Figure 3(IV)] and 30–32 nm [372 nm peak in Figure 3(III)]. This indicates that the transverse modes appeared at significantly shorter wavelengths in NWs with pentagonal cross sections than in the abovementioned wires. Besides causing a blue shift in the peaks, reducing the NW diameter reduces the amount of scattered light.

STEF films based on a 15-nm-diameter Ag NW networks embedded into cross-linked PDMS elastomer were formed via a conventional spin-coating technique that adhered the NWs to a substrate, as shown in Figure 1. Ag NW were dispersed in DI water at a density of 0.2 mg/mL and directly coated onto the Si wafer substrate, which was previously cleaned with acetone, following by drying at 80 °C for 5 min. Second, 0.01 wt% silica gel dispersed in ethanol was spin-coated at 1000 rpm, and post drying, liquid PDMS with a thickness of ~50 µm was coated on the upper surface of silica and the Ag NWs network layer, followed by curing and crosslinking. Afterwards, we peeled the cured PDMS from the Si wafer. Here, when liquid PDMS covers the Ag NW network layer, it penetrates into the interconnected pores of the Ag NW network because of its low viscosity and low surface energy. After curing, all Ag NW networks are buried on the cross-linked PDMS surface (crosslinking between PDMS and silica gel) without considerable voids, indicative of the successful transfer of Ag NW networks from Si wafers to PDMS and excellent adhesion between Ag NW and PDMS. The Ag NW network is embedded into the surface of ~50-µm-thick-PDMS films.

Figure 4(I) shows a photograph of the finally produced Ag NW network embedded into the PDMS film sample, which is STEF, and Figure 4(II) and (III) shows the SEM and AFM surface images of the Ag NW conductive network layer, respectively. In particular, the SEM image of Figure 4(II) shows a highly transparent, extensible, and reliable “STEF” based on a 15-nm-diameter Ag NW network layer embedded in the surface layer of the cross-linked PDMS elastomer film. Here, PDMS completely penetrated into the Ag NW network and filled the gaps between Ag NWs, as shown in the SEM surface image of Figure 4(II), affording an Ag NW network and PDMS. The Ag NW network structure embedded in the surface layer of the cross-linked PDMS elastomer film was clearly observed as the current map image of AFM in Figure 4(III). The sheet resistance and optical value of the Ag NW conductive network layer was determined as a function of density of the Ag NWs in the network. That is, the change in the sheet resistance with increasing density (the content of the Ag NW networks in the layer is described by areal density, namely the Ag NW weight per unit area of the films) of the Ag NW network layer is obtained. However, the sheet resistance of the Ag NW network layer significantly decreased with increasing Ag NW density. Their results showed a low sheet resistance of 20, 40, 50, and 85 Ω/sq at transmittances of 90%, 95%, 96%, and 97% (based on PDMS), respectively. In particular, these 15-nm-diameter Ag NW network embedded PDMS films were exhibited to have low haze values of less than 1.5% (net Haze) with a transparency of 90% despite the low sheet resistance of 20 Ω/sq (up to 90% transmittance and ~1% haze at the sheet resistance of 60 Ω/sq). These haze values shown above were approximately 0.2–0.3 lower at the same sheet resistance condition than that of the 20-nm-diameter Ag NWs reported in the previous work [23]. However, as the diameter is decreased, the optical haze parameter improved; thus, the scattered light can be reduced and the haze value is greatly decreased. As a result, it has been suggested that a 2-D percolating network film constructed using at least 15-nm-diameter Ag NWs is needed to satisfy the electrical and optical properties of crystalline ITO glass.

Elastic behavior was observed for the sample under dynamic loading. In Figure 5(I), at tensile strains of 10%, 20% and 30%, the change in R/R0 was observed at strain restoration for the tensile strain. The initial sheet resistance (R_0_) was almost completely recovered for a stretch/release cycle test with strains ε of 10% and 20%, revealing the outstanding stretchable property of film. Nevertheless, at a strain of greater than or equal to 30%, the sheet resistance of the film was not restored to its original position. Given that the flexible and stretchable characteristics of Ag NW network-embedded PDMS film can obtain highly reliable mechanical performance under continuous strain deformation, repeated stretch/release tests were conducted on the films. An automated testing tool was utilized, which enabled the electrode to exhibit repeated alternate stretch and release. This repeated stretch and release led to cyclic fatigue failure. Thus, the resistance of the Ag NW network-embedded PDMS film sharply increases at the very first stretching and then returns to its initial value. In this test, elongation values of 10% and 20% were utilized. With the repetition of the test for ≥100 cycles under stretching conditions of 10% and 20%, the change in the resistance was restored to its original position without any change in the resistance (Figure 5(II)). However, highly stretchable films based on the 15-nm Ag NW networks embedded into the cross-linked PDMS elastomer were simply fabricated using a spin-coating method. 

## 3. Conclusions

In conclusion, we demonstrated for the first time that ultra-fine Ag NWs with 15-nm-diameter that could not be realized in previous work [23] via a pressure-induced polyol process and in the presence of AgCl with KBr and K^+^ ions induced a notable pressure effect. The characteristic SPR of these 15-nm-diameter NWs appeared at 362 nm. This is a novel finding for Ag NWs and provides evidence of their high optical performances. Furthermore, we fabricated the stretchable transparent electrode films (STEF) based on a 15-nm-diameter Ag nanowires networks embedded into a cross-linked polydimethylsiloxane elastomer. These 2-D embedded Ag NW network film with a 15-nm-diameter Ag NW showed a low sheet resistance of 20 Ω/sq. at 90% transparency with haze values (<1.5%). The electrode films also exhibited a high elasticity of 20%, and the strain films exhibited a good response to the stretch/release of 100 cycles and hysteresis tests. However, these 2-D STEF exhibit good flexibility, making them promising candidates for use as a transparent electrode in flexible electronics. In particular, in the case of the Ag NW embedded elastomer films having a high stretchability and a high electric conductivity, as in the present study, it is expected that these films will provide a much higher performance material in many areas for flexible transparent devices that can replace ITO.

## Figures and Tables

**Figure 1 nanomaterials-09-00576-f001:**
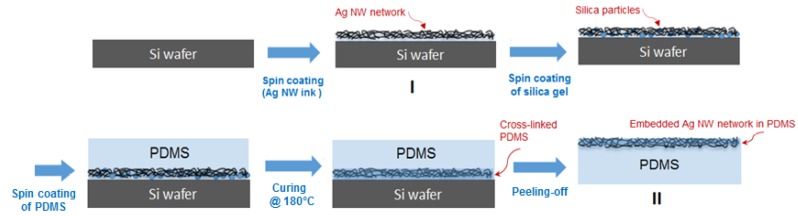
Fabrication of highly strain electrode films based on a Ag NW networks embedded into the cross-linked PDMS elastomer.

**Figure 2 nanomaterials-09-00576-f002:**
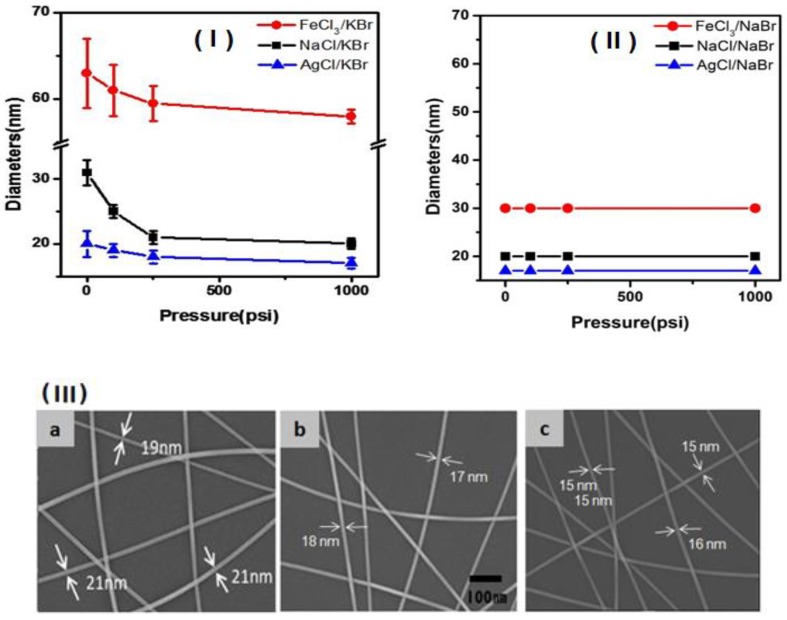
Ag NW diameter vs. pressure in the presence of various salts: [**I**] AgCl–KBr, NaCl–KBr, and FeCl_3_–KBr and [**II**] AgCl–NaBr, NaCl–NaBr, and FeCl_3_–NaBr (the error range is observed within the range of 2~3 nm, respectively). [**III**] SEM images of the Ag NWs synthesized in the presence of AgCl–KBr salts; (**a**) 19–25, (**b**) 17–18, and (**c**) 15–16 nm, respectively. These NWs correspond with 0, 250, and 1000 psi (69 bar), respectively.

**Figure 3 nanomaterials-09-00576-f003:**
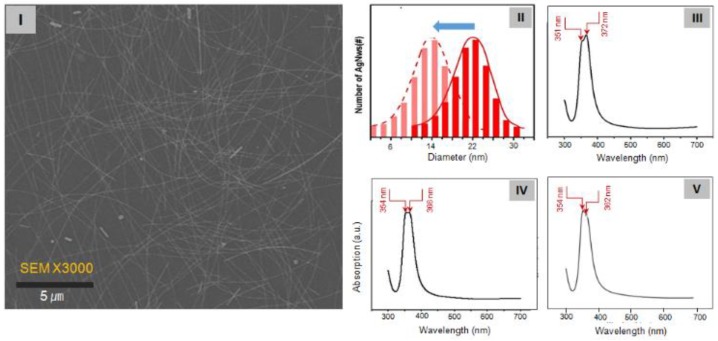
[**I**] SEM images of Ag NWs at low magnification (3000×). The average diameter of the NWs is 15 nm. [**II**] Diameter distribution of the Ag NWs synthesized at 1000 and 0 psi. Surface Plasmon resonance (SPR) absorption characteristics of the synthesized Ag NWs with diameters of [**III**] 30–32 nm, [**IV**] 20–22 nm, and [**V**] 15 nm.

**Figure 4 nanomaterials-09-00576-f004:**
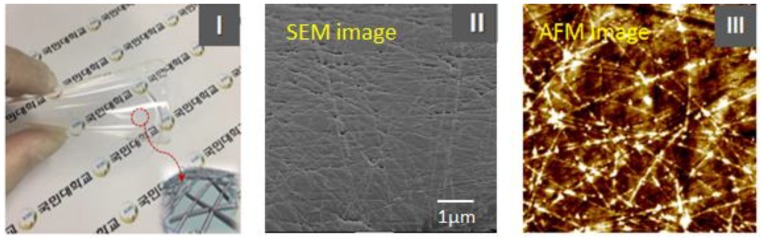
[**I**] Photograph of a STEF sample and [**II**] surface SEM image and [**III**] AFM current image of the Ag NW network-embedded PDMS.

**Figure 5 nanomaterials-09-00576-f005:**
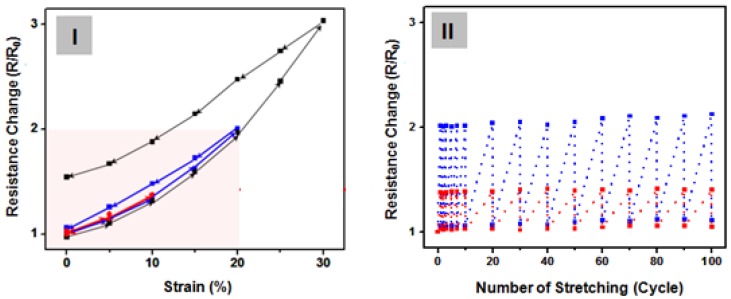
(**I**) Hysteresis curve of the film comprising a Ag NW-network-embedded PDMS film (at tensile strains (ε) of 10%, 20%, and 30%). (**II**) Effect of repeated stretching on the resistance change (R/R_0_) at strain recovery (stretch/release cycles of ε = 10% and 20%).

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
