# Peer review of "Stretchable and Low-Haze Ag-Nanowire-Network 2-D Films Embedded into a Cross-linked Polydimethylsiloxane Elastomer"

_nanomaterials, 2019, doi:10.3390/nano9040576_

Reviewer 1 Report

The authors developed stretchable transparent conductive films with nm diameter Ag nanowires networks in PDMS. The NWs were synthesised using pressure-induced polyol with AgCl particles. The films were transparent and presented remarkable sheet resistance of 20 Ohm/sq. Elasticity was as high as 20%.

The paper is strong and has been well written

Just a few minor comments

1-     Other application on silver NPs in PDMS such as gas separation: Journal of Membrane Science 470, 346-355, 2014, and heat transfer through PDMS Lab on a Chip 14 (17), 3419-3426, 2014

2-     Change “we newly report a pressure-induced” to “we report a novel pressure-induced”

3-     Figure 4 II is not clear – please redo it and remove the bottom information bar

4-     Add to the discussion in the conclusion  

Author Response

 We have inserted the following four comments into the manuscript by item.
 1-     Other application on silver NPs in PDMS such as gas separation: Journal of Membrane Science 470, 346-355, 2014, and heat transfer through PDMS Lab on a Chip 14 (17), 3419-3426, 2014

2-     Change “we newly report a pressure-induced” to “we report a novel pressure-induced”

3-     Figure 4 II is not clear – please redo it and remove the bottom information bar

4-     Add to the discussion in the conclusion  

Reviewer 2 Report

A Manuscript entitled "Stretchable and Low-Haze Ag-Nanowire-Network 2-D Films Embedded into a Cross-linked Polydimethylsiloxane Elastomer" is about hot topic of stretchable and low-haze Ag-Nanowire containing PDMS elastomer based electronically conductive composites. 

There are many similar publications in this area, but this manuscript introduces significant progress and therefore deserves to be published. The manuscript is well written and, in my opinion, deserves to be published in its current form.

Author Response

We have slightly supplemented and re-submitted according to the viewer's comment.

Reviewer 3 Report

I would recommend this paper to be accepted. I have only one minor concenr about the references formatting. Once you give title of the cited paper, once not. Please unify it and make according tot he MDPI Nanomaterial's standard.

Author Response

(The authors gave the same response as above.)
